# Development of 3D Microstructures for the Formation of a Set of Optical Traps on the Optical Axis

**Dmitry A. Savelyev** [1,2,*] and **Sergey V. Karpeev** [1,2,*]

1 Samara National Research University, 443086 Samara, Russia
2 Image Processing Systems Institute of RAS—Branch of the FSRC "Crystallography and Photonics" RAS, 443001 Samara, Russia
* Correspondence: dmitrey.savelyev@yandex.ru (D.A.S.); karp@ipsiras.ru (S.V.K.)

**Abstract:** Three-dimensional (3D) microstructures used in the formation of optical traps on the optical axis in the near diffraction zone are calculated and studied. Subwavelength, variable-height annular gratings (a lattice period of $1.05\lambda$) with a standard and graded-index (GRIN) substrate are considered as microstructures. Two scenarios are examined for changing the refractive index $n$ of the GRIN substrate: from a maximum $n$ in the center to a minimum $n$ at the edges (direct GRIN) and, conversely, from a minimum $n$ in the center to a maximum $n$ at the edges (reverse GRIN). The propagation of light through the proposed 3D microstructures is simulated using the finite-difference time-domain (FDTD) method. The possibility of obtaining not only single but also a set of optical traps on the optical axis is demonstrated. It is also shown that compared to the results obtained with a diffractive axicon, the size of the focal spot can be reduced by 21.6% when use is made of the proposed 3D microstructures and the light needle is increased by 2.86 times.

**Keywords:** optical vortices; GRIN; optical traps; 3D microstructures; FDTD; meep

## 1. Introduction

An urgent task of modern research is the trapping and manipulation of micro-objects in various media [1–11]. Optical tweezers ("optical traps") [12–14] allow microscopic objects to be manipulated using the laser light. The invention of optical tweezers made it possible to advance qualitatively and quantitatively in biological and biophysical research, especially in studies of the behavior of single molecules [15–18], including changes in the position of complex three-dimensional (3D) biological objects [19]. Note that with the technological improvement of modern optical and electronic devices, the limits of applicability of previously developed methods are expanded [2,3,5,6]. An important problem of modern studies is the further improvement of the method of optical tweezers and its various configurations [20–26], including optoelectronic tweezers [22], optothermal tweezers [23], and optical manipulation on solid substrates [24]. In particular, it has been shown that a new kind of optically controlled nanotweezers (opto-thermo-electrohydrodynamic tweezers) enables the trapping and dynamic manipulation of nanometer-scale objects at locations that are several micrometers away from the high-intensity laser focus [20]. A new type of light-based tweezers (opto-refrigerative tweezers) was shown to exploit solid-state optical refrigeration and thermophoresis to trap particles and molecules at the laser-generated cold region [21]. It should also be noted such methods are suitable for versatile manipulations as holographic tweezers [25] and DMD-based optical manipulation [26].

In many applied problems of optical trapping, 3D optical traps are used, which represent a region with a minimum light intensity, uniformly surrounded by intensity maxima ("optical bottle") [4,27,28]. A 3D trap is known to be implemented using optical fields of various configurations, including Hermite–Gaussian beams and Laguerre–Gaussian modes [29–32]. Such 3D optical structures can be obtained using axisymmetric diffractive structures, including axicons [33–37].

Generally, the special properties of vortex beams have been actively studied in recent decades [30,38–42]. One of the most important properties is the orbital angular momentum [43,44], which is determined by the order of the optical vortex (topological charge). A distinctive feature of an optical vortex beam is the presence of vortex phase singular points at which the phase is not defined, and the amplitude is zero. Along with optical trapping and manipulation [45,46], vortex beams are used for optical information transmission [47,48], laser processing [49,50], probing [51], and control of ultrafast optical field using ultrashort pulses [52].

As is well known, structured vortex beams can be generated using various elements of diffractive optics, such as spiral and twisted axicons [53–55], spiral phase plates [56–60], and multi-order diffractive optical elements [61–65]. Thus, spiral phase microplates on an optical fiber endface were used to form vortex beams in optical connector systems in data centers [66], and a miniature (less than 200 μm in size) integrated mode sorter (a converter of the vortex phase into an inclined plane wavefront) made it possible to demultiplex vortex beams in a wide spectral range (up to 300 nm) [67].

Graded-index (GRIN) materials are currently actively used for a number of applications [68–72], including collimation of light [73,74], micro-optical communication [69,75], and biomedicine [76–78]. As a rule, graded elements are to some extent an analogue of a lens that forms a short focus; therefore, it is logical to use them to couple optical fibers for information transmission [79,80]. Then, one element will be used at the fiber output for laser beam scattering and the other will be employed at the fiber input for collecting the laser beam [81]. It is also known that graded elements can control the propagation of light (in particular in integrated photonic chips [82]) for tight focusing [82].

In this paper, we study diffraction of Laguerre–Gaussian (1,0) modes on subwavelength, variable-height optical microelements with a standard and GRIN substrate, which are used to form optical traps on the optical axis. The propagation of laser light (3D) is calculated numerically by the finite-difference time-domain (FDTD) method using high-performance computer systems [83–85]. The calculations are performed on a computational cluster with a capacity of 850 Gflop.

## 2. Materials and Methods

As the input laser light, we used the Laguerre–Gaussian (0,1) mode with a beam width $\sigma = 1.5$ μm:

$$GL_{01}(r, \varphi, z) = \left( \frac{\sqrt{2}r}{\sigma(z)} \right) \exp[ikz - i2\eta(z)] \exp\left[ \frac{i\pi r^2}{\lambda R(z)} \right] \exp\left[ -\frac{r^2}{\sigma^2(z)} \right] \exp(i\varphi), \quad (1)$$

where $\sigma(z)$ is the effective beam radius, $z_0 = \pi\sigma_0^2/\lambda$ is the confocal parameter, $\lambda$ is the laser light wavelength, $r^2 = x^2 + y^2$, $\varphi = \text{arctg}(y/x)$, $\eta(z) = \text{arctg}(z/z_0)$, and $R(z) = z(1 + z_0^2/z^2)$ is the radius of curvature of the parabolic front of the light field.

An optical vortex as an input uniformly polarized beam significantly changes the focal pattern compared to a conventional Gaussian beam, and the direction of circular polarization rotation becomes important. It should be noted that for circular polarization in which the sign of circular polarization is opposite to that of the introduced vortex phase singularity (left-handed circular polarization), an intensity peak is observed in the center. When the sign of circular polarization is co-directed with the introduced vortex phase singularity (right-handed circular polarization), a shadow round light spot is formed (i.e., an area with a minimum intensity in the center) [41,86,87]. At the second order of the optical vortex and higher, a shadow round light spot is formed for left-handed circular polarization [88]. An optical trap is formed by intensity fluctuations on the optical axis, and so we will consider the first order of the optical vortex in the incident beam to obtain an intensity peak. This will allow us to further select the design of 3D elements so as to form optical traps on the beam propagation axis.

The finite-difference time-domain (FDTD) method is one of the most common techniques for solving Maxwell's equations. With this method, Maxwell's equations are solved by discretization using central differences in time and space; next, they are solved numerically [89]. In this work, the solution was simulated numerically with the software package Meep, which is often used for FDTD simulations [90–92].

Meep was developed by a research group of scientists from the Massachusetts Institute of Technology (MIT). There are two ways to describe models in Meep: either a simple scripting language (Scheme) or a more flexible programming language (C++) [83]. In our work, we used the Scheme programming language and a standard Yee's mesh.

We used the following simulation parameters: The total size of the 3D computational domain together with the PML absorbing layer is 13.5λ. The wavelength of the light is λ = 0.532 μm. The thickness of the absorbing PML layer is 0.55 μm. The sampling step in space is λ/25 and the step in time is λ/(50c), where c is the speed of light. The element substrate was partially immersed in the PML.

Subwavelength, variable-height annular gratings (a lattice period of 1.05λ) with standard and GRIN substrates were used as elements. The refractive index n of the GRIN substrate varied from 1.47 to 2.7 (Figure 1). The results were compared with those obtained with a diffractive axicon having the same period (i.e., with a numerical aperture NA = 0.95) and a standard substrate at n = 1.47. The relief refractive index for all elements was 1.47.

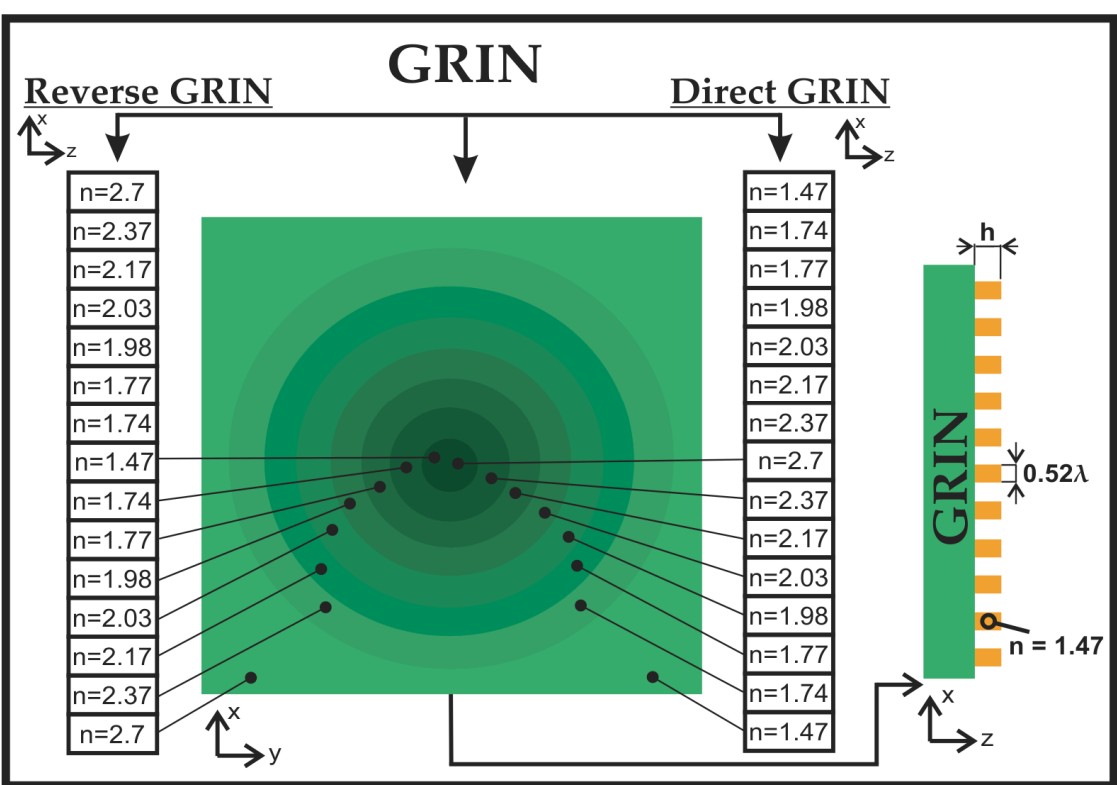

**Figure 1.** Appearance of direct and reverse GRIN substrates and cross sections *xy* and *xz*.

The relief height *h*, corresponding to a phase jump π radians at *n* = 1.47, is calculated by the formula:

$$h = \frac{\pi}{k(n-1)} = \frac{\lambda}{2(m-1)} = 1.06382\lambda \approx 1.06\lambda, \tag{2}$$

We investigated the change in the height of individual rings of the considered gratings with a step *S* corresponding to a phase jump of π or π/2 radians, that is, 1.06λ and 0.53λ.

Two scenarios of changing the refractive index of the GRIN substrate were considered: from a maximum *n* in the center to a minimum *n* at the edges (direct GRIN), and vice versa,

from a minimum $n$ in the center to a maximum $n$ at the edges (reverse GRIN). Figure 1 shows cross sections of the studied GRIN substrates in the $xy$ plane.

Propagation through a standard substrate and direct and reverse GRIN substrates without relief are shown in Figure 2, where their effect is noticeable: a direct GRIN substrate has a converging lens effect, while a reverse GRIN substrate has a diverging lens effect. The thickness of the substrates in question was fixed and equal to $\lambda$.

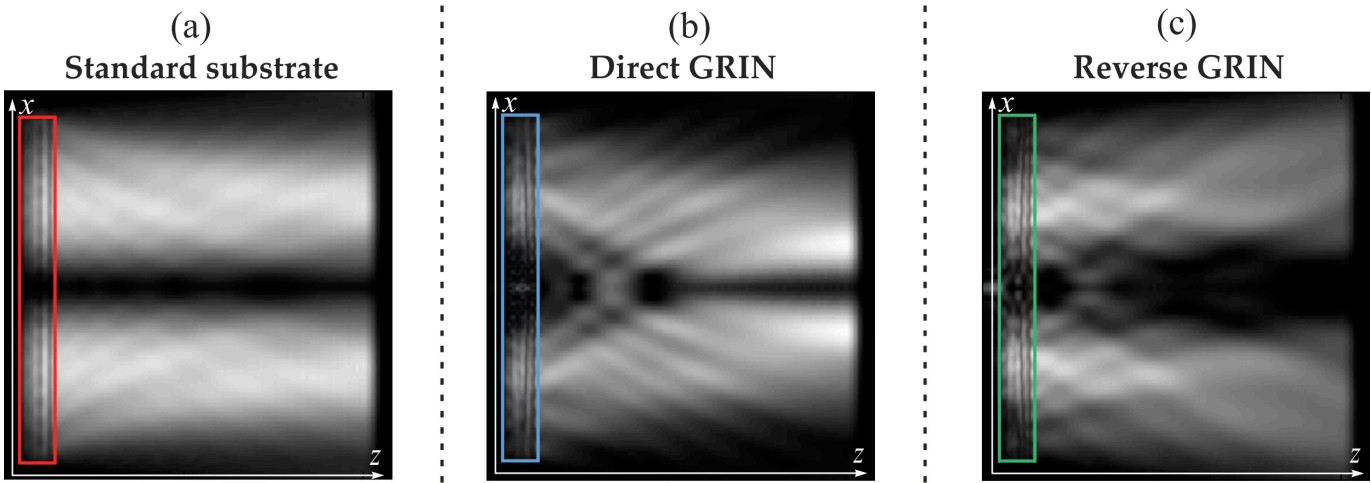

**Figure 2.** Optical vortex diffraction on (**a**) a standard substrate (red), (**b**) a direct GRIN substrate (blue), and (**c**) a reverse GRIN substrate (green), $xz$ cross sections, intensity.

## 3. Results

In this section, we will consider in detail the focusing of vortex laser beams by the 3D microstructures that are used in the formation of optical traps on the optical axis.

### 3.1. Development of 3D Microstructures

As was previously shown, a change in the height of a diffractive axicon significantly affects the diffraction pattern in the near zone of the element [40,41,93–99], and in the case of uniformly polarized light, a high-power longitudinal component of the electric field can be formed due to the energy exchange between the components of the electromagnetic field [41,100,101].

In this work, subwavelength, variable-height annular gratings (a lattice period of 1.05$\lambda$) were considered as microstructures. As previously mentioned, the heights of individual relief rings were varied with a step $S$ equal to 1.06$\lambda$ and 0.53$\lambda$. Thus, we considered the following heights of the rings: 1.06$\lambda$, 2.12$\lambda$; 3.18$\lambda$; 4.24$\lambda$; 5.3$\lambda$; 6.36$\lambda$ (for $S$ = 1.06$\lambda$) and 1.06$\lambda$; 1.59$\lambda$, 2.12$\lambda$; 2.65$\lambda$; 3.18$\lambda$; 3.71$\lambda$ ($S$ = 0.53$\lambda$).

Figures 3–5 show cross sections of the optical elements in question. They were compared with a diffractive axicon having a numerical aperture of 0.95. We also considered direct (the height of the relief varies from a maximum in the center to a minimum at the edges) and reverse GRIN substrates (the height of the relief changes from a minimum in the center to a maximum at the edges).

When changing the relief height, we considered not only a standard substrate but also direct and reverse GRIN substrates (Figure 4).

Let us fix the height of the central part of the ring, as well as the height of the subsequent alternate rings (height $h_1$ = 1.06$\lambda$).

At the same time, the heights of the remaining rings (height $h_2$) will be changed, i.e., we will vary the heights of not all but only of even zones of optical elements. Examples of annular gratings are shown in Figure 5.

The following subsections will show the results of the FDTD simulation for the elements considered in this subsection.

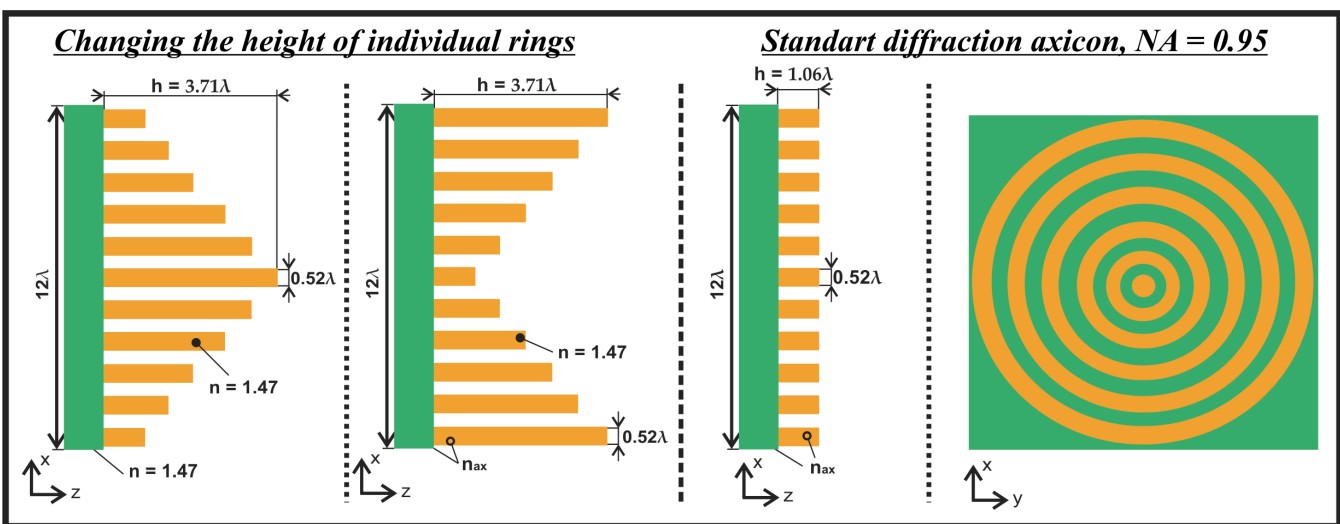

**Figure 3.** Cross sections of the studied 3D microstructures with a standard substrate in *xz* and *xy* planes.

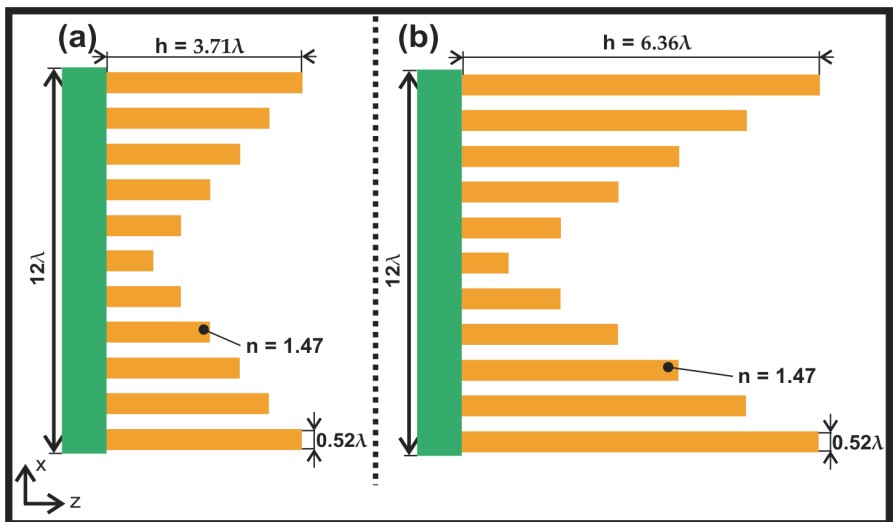

**Figure 4.** Cross sections of the studied 3D microstructures with a GRIN substrate in the *xz* plane.

### 3.2. Formation of Single Optical Traps by Annular Gratings

In this subsection, we study the diffraction of the Laguerre–Gaussian (0,1) modes with a beam width $\sigma = 1.5$ μm (left-handed circular polarization) on the 3D structures considered in the previous subsection. The results of the studies for the optical elements shown in Figure 3 (i.e., for a diffractive axicon, direct and reverse annular gratings with a standard substrate) are shown in Figure 6.

The size of the focal spot on the optical axis was estimated from the full width at half maximum (FWHM) and the depth of focus (DOF), i.e., the size of the longitudinal light segment was also estimated from the FWHM. All figures below show FWHM values at a maximum on the optical axis.

Note that for a direct annular grating, a maximum is formed inside the element, and outside it, the intensity on the optical axis decreases. The DOF estimate of the light segment was taken for this case inside the element. The focal spot is wider than that for the diffractive axicon; therefore, FWHM and DOF values for the diffractive axicon will serve as a reference for comparison in subsequent studies.

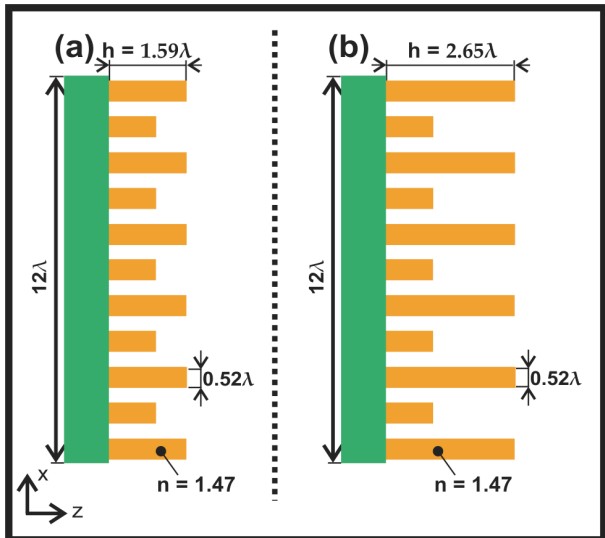

**Figure 5.** Cross sections of the studied 3D microstructures with a GRIN substrate at $h_2 = $ (**a**) 1.59$\lambda$ and (**b**) 2.65$\lambda$.

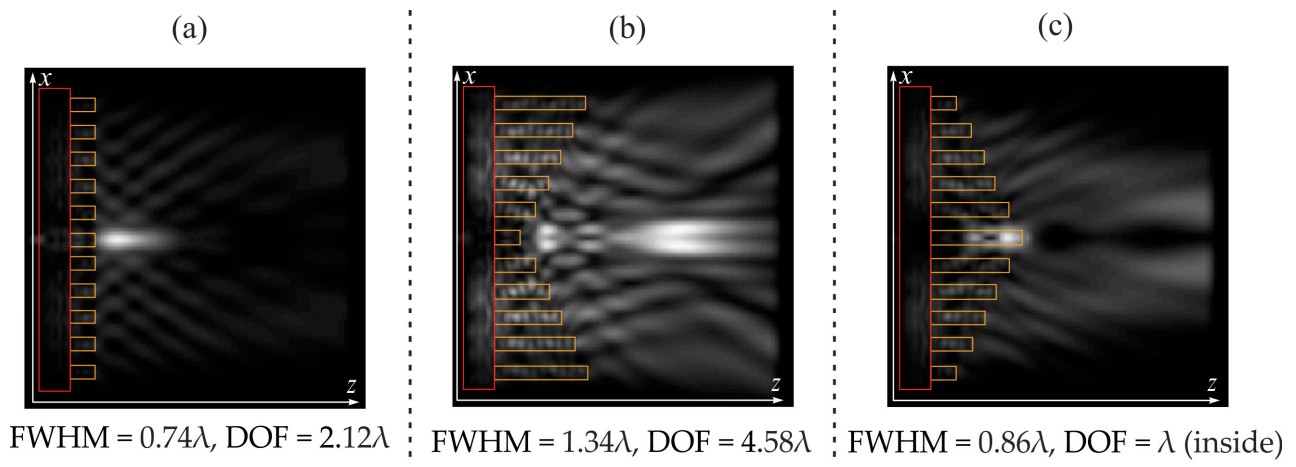

FWHM = 0.74$\lambda$, DOF = 2.12$\lambda$    FWHM = 1.34$\lambda$, DOF = 4.58$\lambda$    FWHM = 0.86$\lambda$, DOF = $\lambda$ (inside)

**Figure 6.** Two-dimensional diffraction pattern of the Laguerre–Gaussian mode (0,1) on variable-height annular gratings in the *xz* plane, total intensity: (**a**) standard diffraction axicon with a height $h = 1.06\lambda$, (**b**) reverse annular grating with a height *h* from 1.06$\lambda$ to 3.71$\lambda$, and (**c**) direct annular grating with a height *h* from 3.71$\lambda$ to 1.06$\lambda$.

For a reverse annular grating, a broadening of the focal spot is also observed with a simultaneous increase in the focal light segment, which is 2.16 times longer than that for a conventional diffractive axicon. An optical trap is also formed at a distance of 3.6$\lambda$ from the central part of the element.

Let us fix the central zone ($h = 1.06\lambda$). Below, we will change the remaining zones.

Our goal is to create an optical trap with a local intensity minimum. An element with alternating steps is a combination of two axicons that give two foci at different distances, which allows one to form a given distribution.

We performed a series of studies with fixed odd zones ($h_1 = 1.06\lambda$) and variable-height even zones, with height $h_2$ lying in the range from 1.06$\lambda$ to 3.71$\lambda$. The formation of optical traps was obtained for heights $h_2$ equal to 1.59$\lambda$ and 2.65$\lambda$. Figure 7 shows longitudinal sections of the propagation intensity of the laser light in question on elements with alternating steps in the case of a standard substrate, direct GRIN substrate, and reverse GRIN substrate.

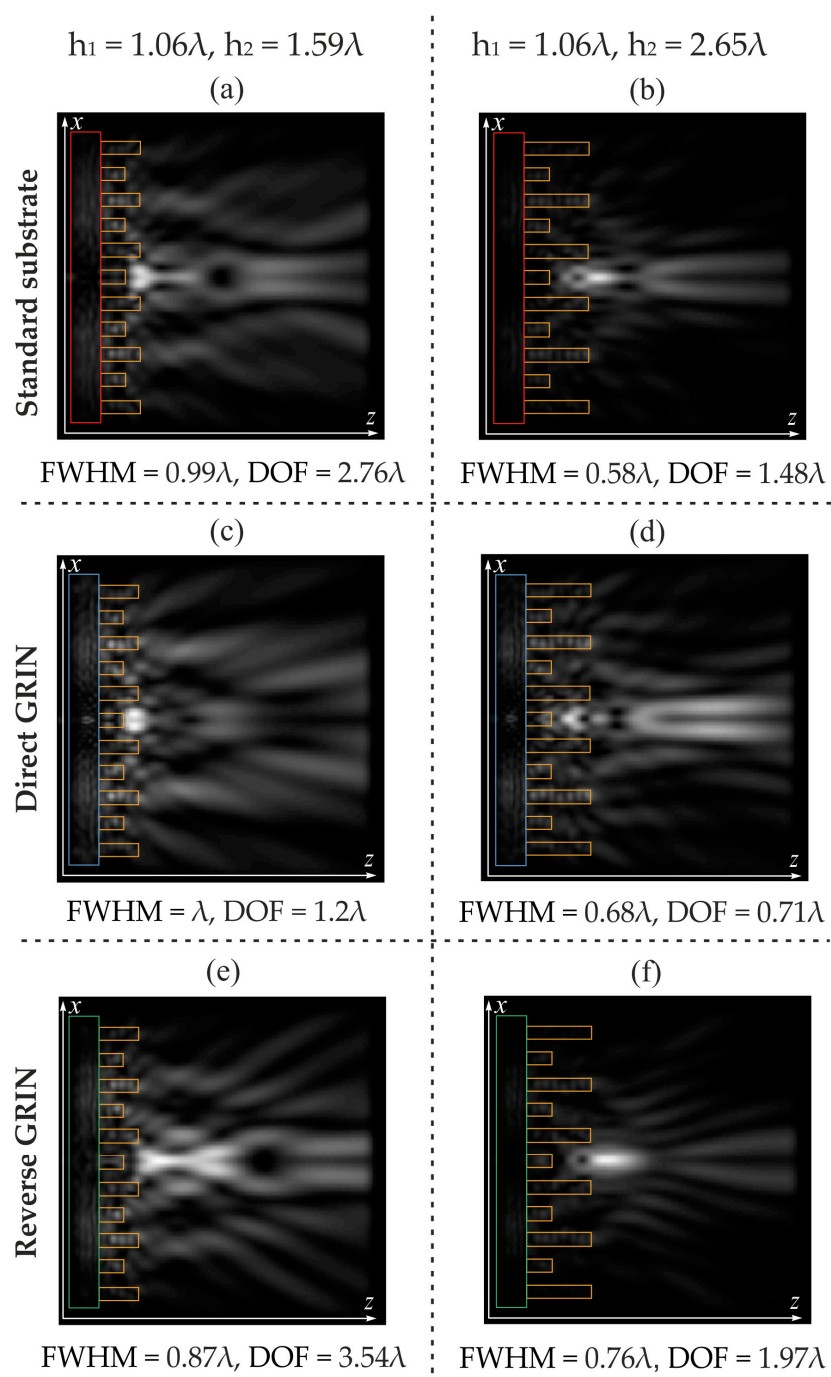

**Figure 7.** Two-dimensional diffraction pattern (intensity, *xz* plane) of the laser light propagation through annular gratings with alternating steps *S* for (**a**,**b**) a standard substrate; (**c**,**d**) a direct GRIN substrate; and (**e**,**f**) a reverse GRIN substrate.

The analysis of Figure 7 shows that for a height $h_2 = 2.65\lambda$, optical traps are formed close to the element for all types of substrates in question, up to an intensity maximum on the optical axis. Moreover, for the case of Figure 7d, the formation of a 'prototype' of a second optical trap is observed with a simultaneous increase in the intensity on the optical axis. The minimum focal spot was obtained for a standard substrate with FWHM = $0.58\lambda$, which is 21.6% less than that of a diffractive axicon.

For a height $h_2 = 1.59\lambda$, optical traps are formed after a maximum at a distance of several wavelengths from the central part of the element. The longest light needle was

obtained for the case of a reverse GRIN substrate with DOF = 3.54λ, which is 1.67 times longer than that formed by a diffractive axicon.

The most pronounced optical trap is formed far from the element in the case of a standard substrate at $h_2$ = 1.59λ (Figure 7a). Let us consider this case in more detail (Figure 8).

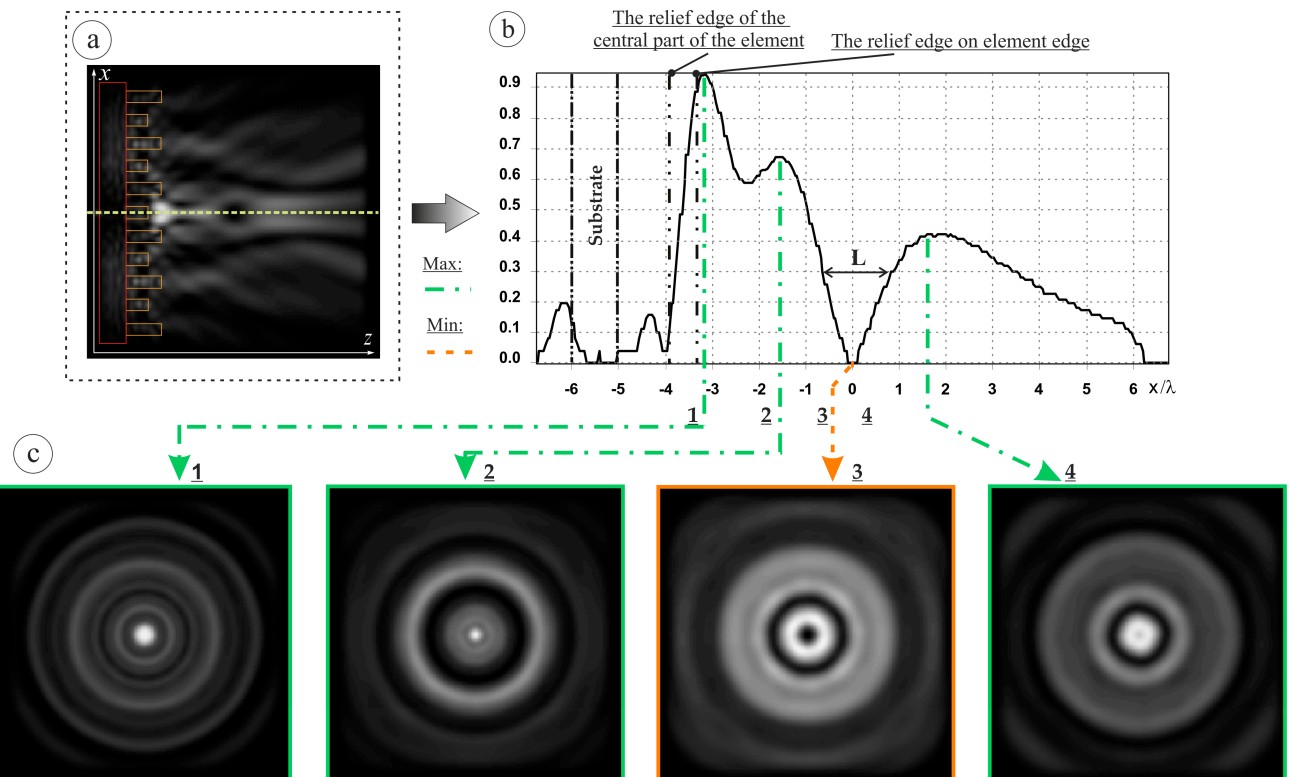

**Figure 8.** Two-dimensional diffraction pattern of the Laguerre–Gaussian mode (1,0) (see Figure 7a): (**a**) *xz* plane, (**b**) cross section in the *xz* plane, and (**c**) cross sections in the *xy* plane.

In the longitudinal plane, the maximum height of the intensity peaks on the optical axis, limiting the optical trap, is 66.7% and 42.2% of the maximum intensity (outside the optical axis). The length of the low intensity region is $L$ = 1.52λ. If we consider the cross section in the region of the minimum (Figure 8), the formation of a ring is observed, with a peak equal to 35.4% of the maximum intensity. The ring width is FWHM = 0.83λ. The width of the shadow focal spot (a minimum in the center) is FWHM = 0.82λ.

We can assume that an increase in the number of combined axicons will form a set of traps along the optical axis of the element. Let us consider below reverse annular gratings with different steps $S$ of height variation.

### 3.3. Formation of Multiple Optical Traps by Reverse Annular Gratings

In this subsection, we study the diffraction of first-order optical vortices on reverse annular gratings of variable height, with the heights of individual relief rings being varied with a step $S$ equal to 1.06λ и 0.53λ.

Figure 9 shows longitudinal sections of the propagation intensity of the laser light in question on reverse annular gratings with a standard substrate, direct GRIN substrate, and reverse GRIN substrate.

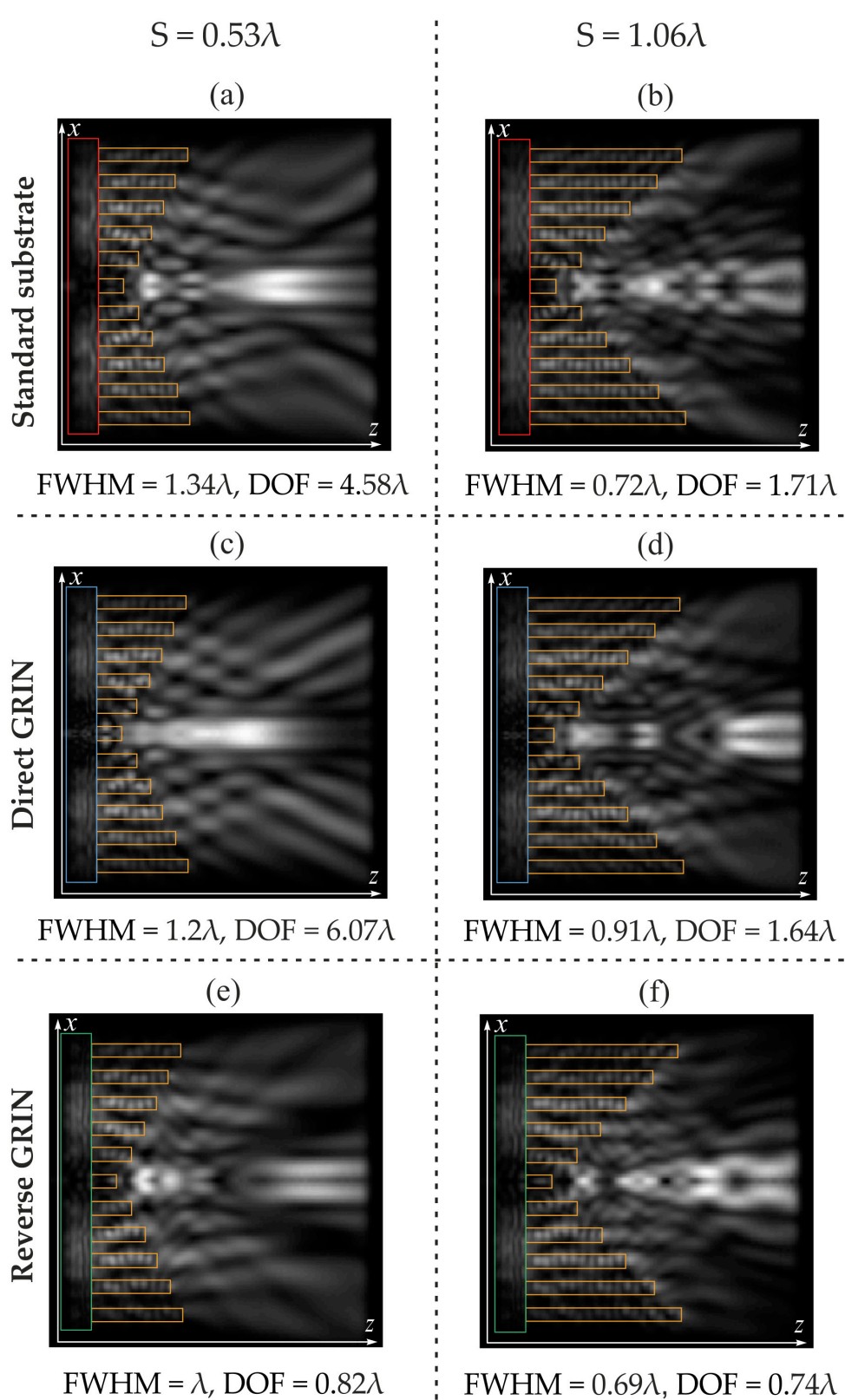

**Figure 9.** Two-dimensional diffraction pattern (intensity, *xz* plane) of the laser light propagation through reverse annular gratings with different steps *S* for (**a**,**b**) a standard substrate; (**c**,**d**) a direct GRIN substrate; and (**e**,**f**) a reverse GRIN substrate.

The analysis of Figure 9 clearly demonstrates the formation of individual optical traps and their sets, which is more typical for a sharper change in height, i.e., for $S = 1.06\lambda$.

For $S = 0.53\lambda$, optical traps are less pronounced; in particular, for the case of a direct GRIN substrate (Figure 9c), a dip on the optical axis is minimal, but the light needle is the longest: DOF = $6.07\lambda$, which is 2.86 times longer than the light needle obtained with a diffractive axicon.

The minimum focal spot was obtained for the case of a reverse annular grating at $S = 1.06\lambda$: FWHM = $0.69\lambda$, which is 6.7% less than the focal spot obtained using a diffractive axicon.

Interesting diffraction patterns were obtained for all substrates at $S = 1.06\lambda$. Let us examine them in more detail in Figures 10–12 (for a standard substrate, a direct GRIN substrate, and a reverse GRIN substrate, respectively).

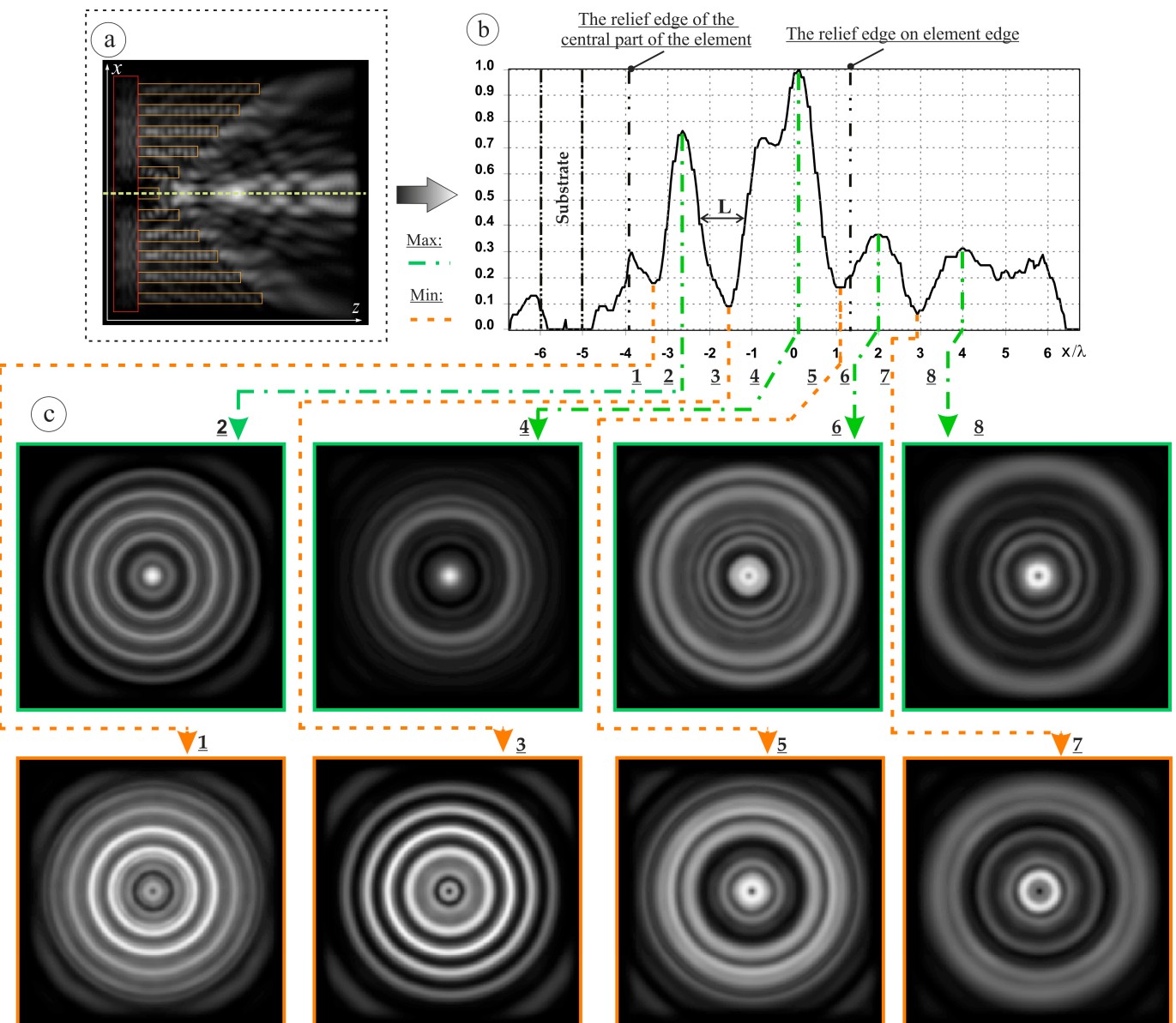

**Figure 10.** Two-dimensional diffraction pattern of the Laguerre–Gaussian mode (1,0) (see Figure 9a,b): (**a**) *xz* plane, (**b**) cross section in the *xz* plane, and (**c**) cross sections in the *xy* plane.

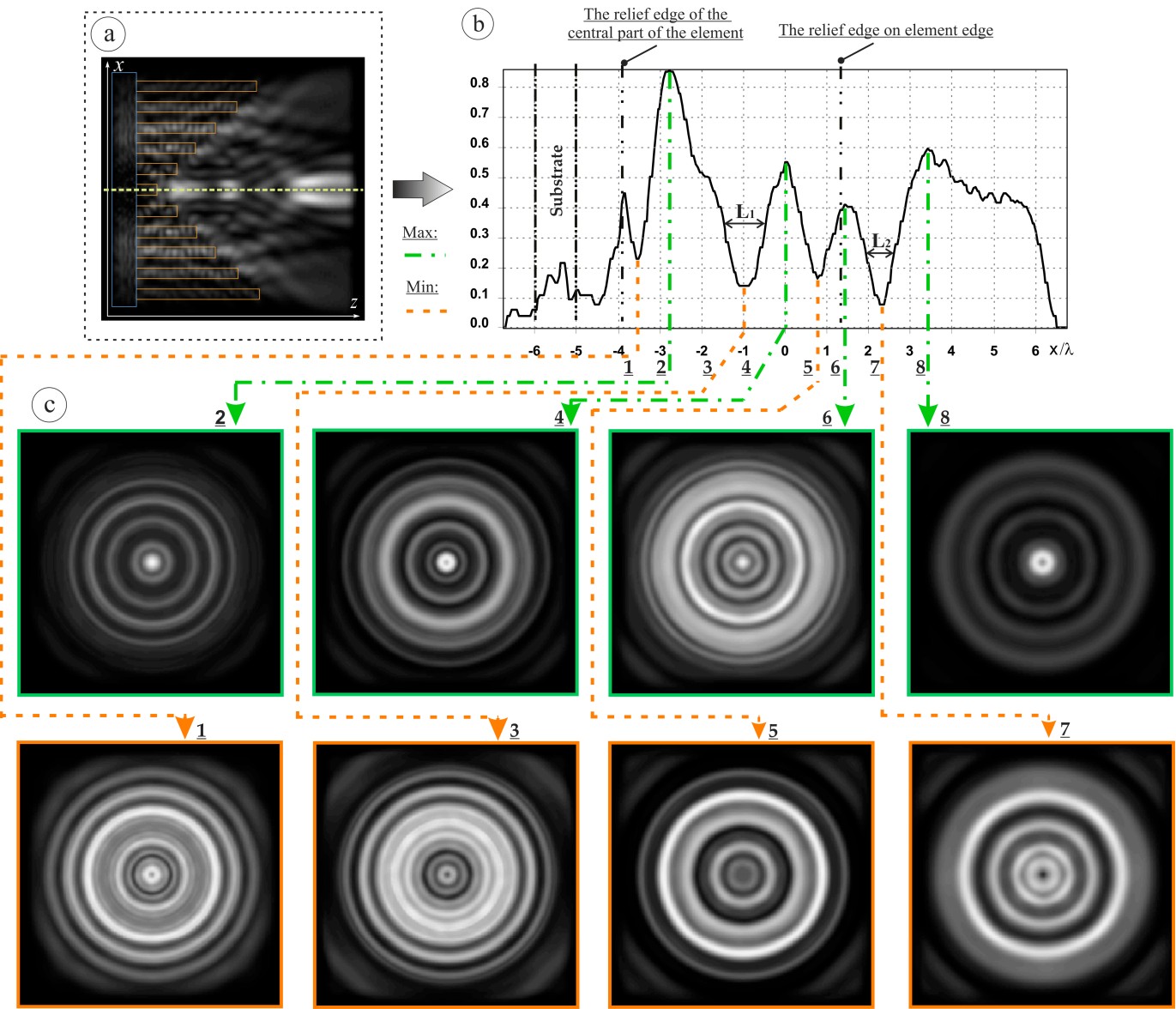

**Figure 11.** Two-dimensional diffraction pattern of the Laguerre–Gaussian mode (1,0) (see Figure 9d): (**a**) *xz* plane, (**b**) cross section in the *xz* plane, and (**c**) cross sections in the *xy* plane.

For a standard substrate (Figure 10), a maximum height of the trap-limiting intensity peaks on the optical axis in the longitudinal plane is 76.1% and 73% (nearest maximum) of the maximum intensity, which is on the optical axis. The length of the low-intensity region for an optical trap is $L = 1.1\lambda$.

A ring is formed, with a peak equal to 33% of the maximum intensity. The ring width is FWHM = 0.45$\lambda$. The width of the shadow focal spot (a minimum in the center) is FWHM = 0.26$\lambda$.

For a direct GRIN substrate (Figure 11), a maximum height of the intensity peaks on the optical axis for the first and second optical traps in the longitudinal plane is 86% and 54.4% and 41.3% and 60% of the maximum intensity (located outside the optical axis in the former case), respectively. The length of the region of reduced intensity for the first trap is $L_1 = 0.93\lambda$, and for the second trap, $L_2 = 0.67\lambda$. Rings are also formed, with the ring for the first optical trap being more pronounced: the peak intensity is 26% of the maximum intensity, the ring width is FWHM = 0.32$\lambda$, and the width of the shadow focal spot (a minimum in the center) is FWHM = 0.25$\lambda$.

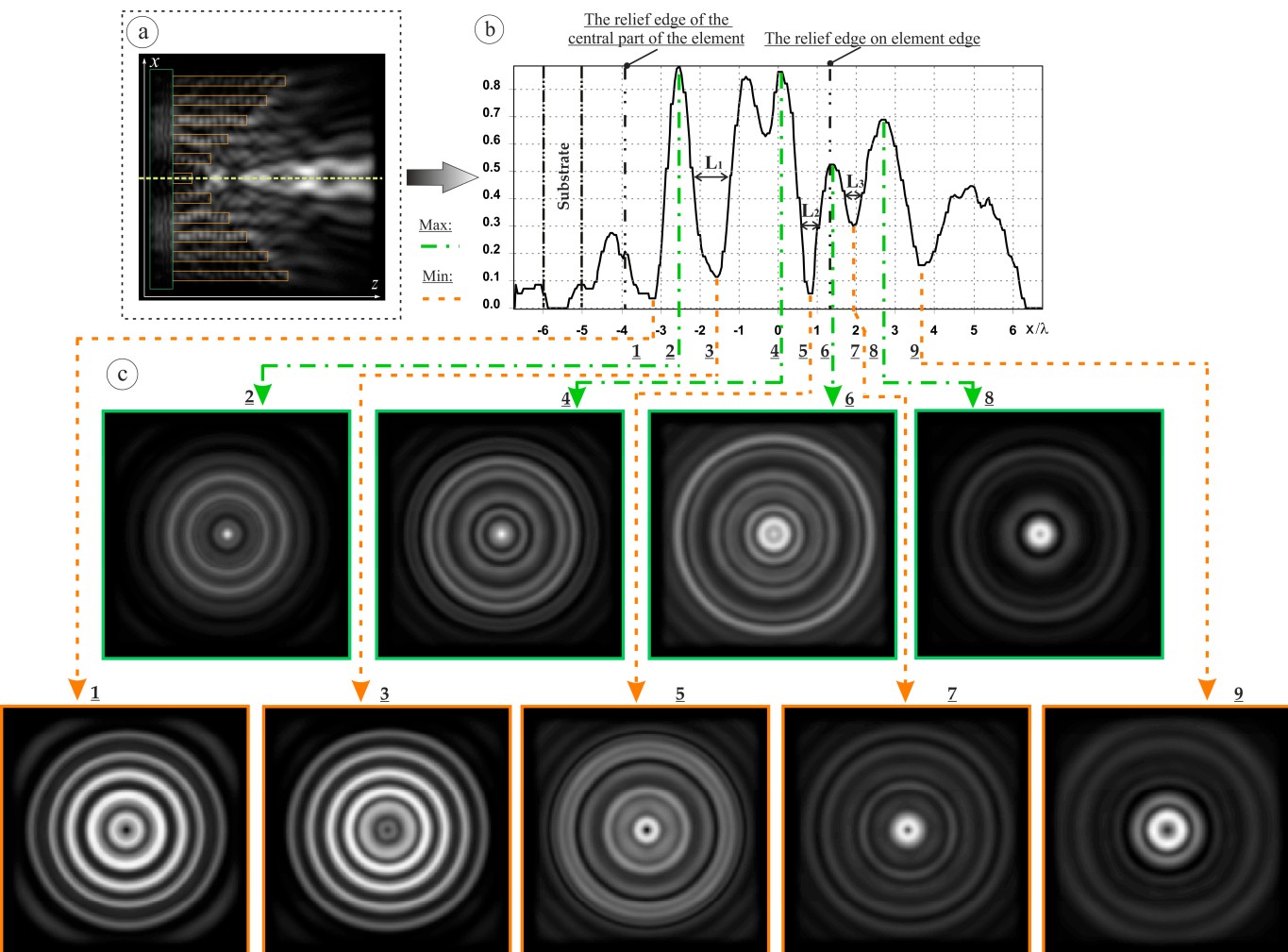

**Figure 12.** Two-dimensional diffraction pattern of the Laguerre–Gaussian mode (1,0) (see Figure 9f): (**a**) *xz* plane, (**b**) cross section in the *xz* plane, and (**c**) cross sections in the *xy* plane.

For a reverse GRIN substrate (Figure 12), a set of three optical traps is observed. The maximum height of the intensity peaks on the optical axis for the first, second, and third optical traps in the longitudinal plane is 87.4% and 83.2%, 84.6% and 53%, and 53% and 69.1% of the maximum intensity (located outside the optical axis in the first case), respectively.

Then, the length of the low-intensity region for the first trap is $L_1 = 0.89\lambda$; for the second trap, $L_2 = 0.48\lambda$; and for the third trap, $L_3 = 0.41\lambda$.

Rings are formed in the planes of the intensity minimum on the optical axis, with peaks equal to 47.2%, 55.6%, and 78.1% of the maximum intensity for the first, second, and third traps, respectively. The ring width of the first, second, and third traps is FWHM = $0.56\lambda$, $0.63\lambda$, and $0.79\lambda$, respectively. The width of the shadow focal spot for the first (a minimum in the center), second, and third trap is FWHM = $2.3\lambda$, $0.34\lambda$, and $0.33\lambda$, respectively.

Thus, when reverse annular gratings are illuminated by left-handed circularly polarized Laguerre–Gaussian (1,0) modes with $\sigma = 1.5$ µm, the formation of a set of relatively narrow (FWHM from $0.25\lambda$ to $2.3\lambda$) and relatively extended ($L$ from $0.41\ \lambda$ to $1.1\lambda$) regions of reduced intensity on the optical axis is observed.

Experiments [10,12,50,102–105] on trapping a particle by a focused laser beam showed the presence of two forces: one force pushes the particle forward along the beam propagation direction, and the other force moves the particle to the region of maximum intensity [50]. The first force is the longitudinal component of the Poynting vector (called scattering), and

the second force is the radiation intensity gradient (called gradient) [50]. However, for the calculation, it is necessary to know not only the field near the particle and calculate the Poynting vector and intensity gradient but also the parameters of the particle and the environment.

All formulas [10,50,105] with sufficient complexity will not explicitly show the action of the trap, as well as the effect on the destruction of the rest of the field. Therefore, let us carry out numerical simulation.

We will consider spheres with a refractive index of n = 1.47 and n = 2.7 as particles. We will vary the radius of these spheres. The simulation results for particles in the first trap are shown in Figure 13, and for the second trap, the results are shown in Figure 14. Spheres with a refractive index of n = 1.47 are shown in yellow; spheres with a refractive index of n = 2.7 are shown in blue.

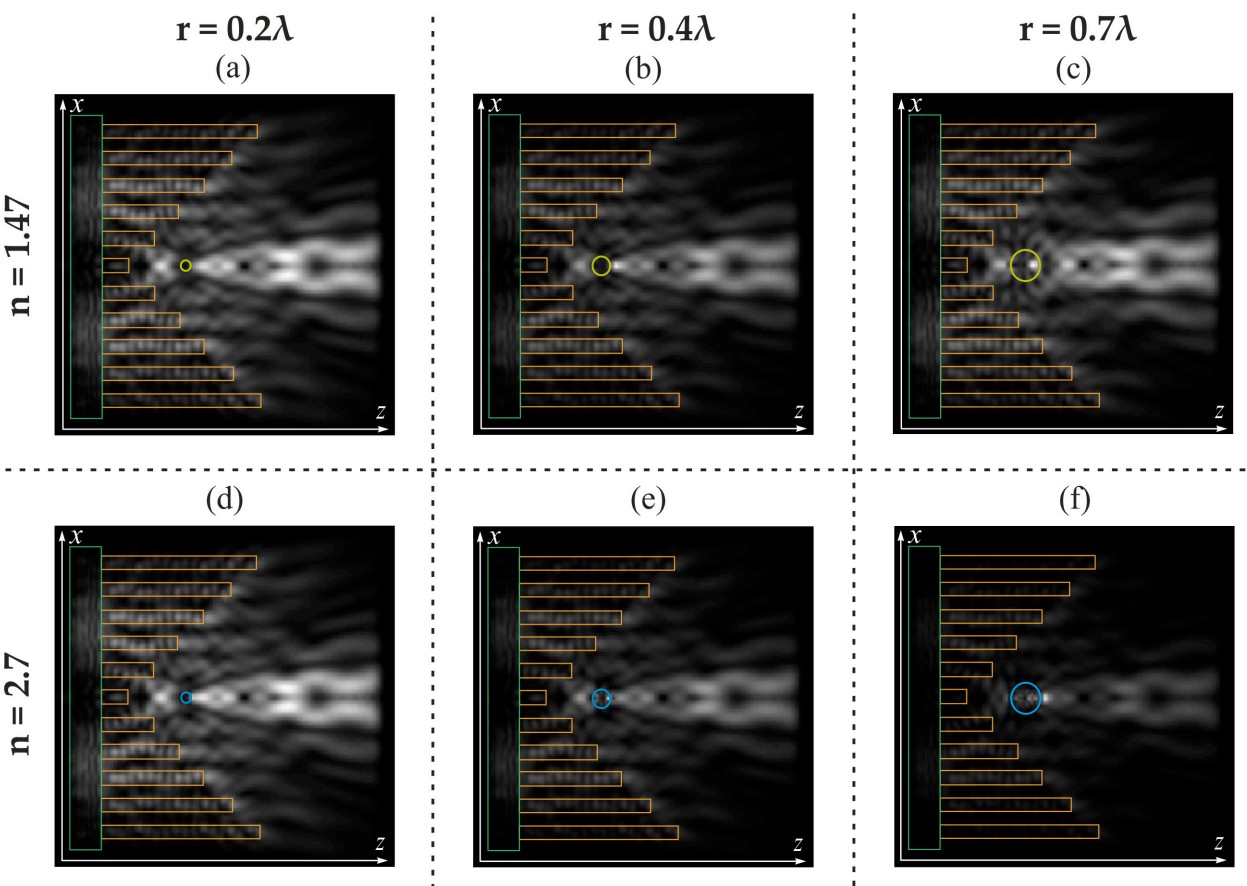

**Figure 13.** Two-dimensional diffraction pattern of the Laguerre–Gaussian mode (1,0) [see Figure 12] with a particle in the first trap: particle with n = 1.47 (**a**–**c**); particle with n = 2.7 (**d**–**f**).

It should be noted that the traps are stable at different particle sizes and refractive indices. Even at a large particle radius (r = 0.7λ), the traps are not strongly destroyed; instead, a decrease in energy is observed.

An increase in the refractive index of the particle leads to a sharper change in the diffraction patterns for both the first and second traps.

It should be noted that for the first trap, at n = 1.47, the intensity maximum is formed outside the sphere, while for n = 2.7, the intensity maximum is shifted to the edge of the sphere. For the second trap, an increase in the particle radius led to the formation of an intensity maximum inside the spheres (see Figure 14).

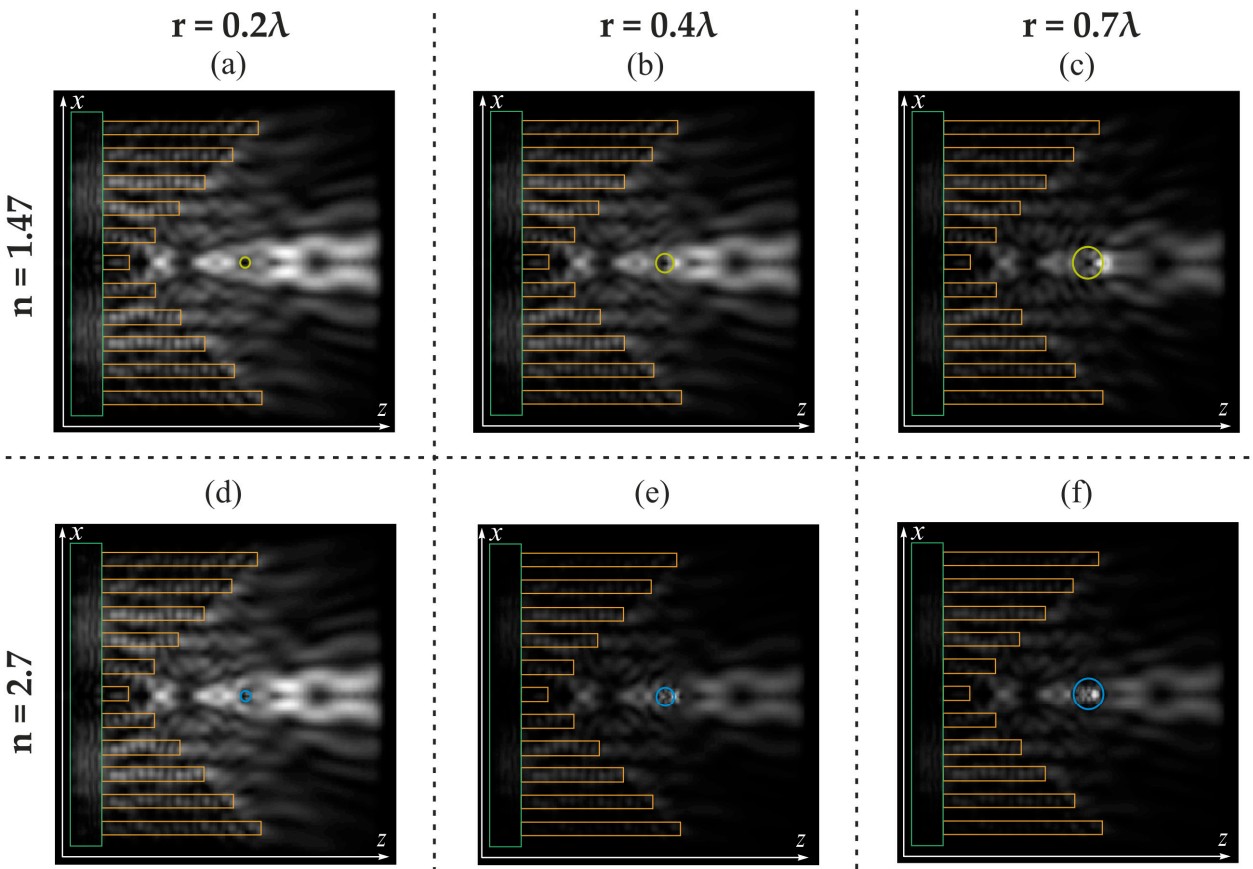

**Figure 14.** Two-dimensional diffraction pattern of the Laguerre–Gaussian mode (1,0) (see Figure 12) with a particle in the second trap: particle with n = 1.47 (**a**–**c**); particle with n = 2.7 (**d**–**f**).

## 4. Discussion

There are various methods for manufacturing GRIN elements. Recently, Yang et al. [106] have demonstrated the possibility of obtaining a spatially varying refractive index, which can arbitrarily vary in three spatial dimensions. Thus, one can speak of free-form GRIN (F-GRIN) media [107,108]. In particular, polymer nanolayers with different refractive indices can be used, which can be combined into single GRIN sheets, and then some desired distribution profile can be formed [109]. Provided that in our case it is not necessary to form any profile, but only a GRIN substrate is required, this method can be also applied.

Note also a possibility of additive manufacturing of GRIN elements [110] when use can be made, for example, of polymer nanoparticles with different (high and low) refractive indices using printing methods. Moreover, additive manufacturing methods have been demonstrated for glass optics with a gradient refractive index [70] where three-dimensional multicomponent green bodies were printed, consisting mainly of silica nanoparticles and various concentrations of titanium dioxide. Then, the green bodies are combined into glass and polished, resulting in optics with individual spatial refractive index profiles [70].

Photopolymerization [111] and direct laser writing [112], in particular for the infrared range [68], are widely known techniques used for the fabrication of GRIN elements.

The types of relief under consideration can be produced by electron lithography; it should also be noted that the possibility of producing a relief with an aspect ratio of more than 100 was previously shown [112–115], and so the production of 3D structures studied in this paper seems possible. It should also be noted that the topographic surface of the structure of the considered elements can lead to the mechanical capture of objects. This is typical for the cases of large particles and the type of relief in the form of reverse annular

gratings or an annular grating with variable-height even zones when there is a difference in heights in the center and the next zone.

Chirality is a term used for systems that lack mirror symmetry. Physical objects like light also have chirality. In general, chiral light carries spin and orbital angular momenta that can be controlled. It is possible to change the chirality of the optical field by changing the left circular polarization to the right circular polarization. In this way, it is possible to create structured light that can be used for optical manipulation [116,117]. In particular, all-solid-phase reconfigurable chiral nanostructures with silicon nanoparticles and nanowires as the building blocks have been proposed for dynamic manipulation of the silicon nanoparticle [116]. It was also possible to create chiral light with an arbitrary angular momentum using a metasurface [117].

## 5. Conclusions

In this work, based on the finite-difference time-domain method, we have studied the focusing features of the circularly polarized Laguerre–Gaussian (1,0) modes on annular gratings with a variable height of the relief rings and a diffractive axicon with standard, direct GRIN, and reverse GRIN substrates. It has been shown that a change in the height of the relief rings of elements markedly affects the diffraction pattern in the near zone: in some cases, a significant decrease in the size of the focal spot, the formation of an extended light segment, and the formation of single and sets of optical traps have been demonstrated. Therefore, the considered elements can have practical application.

An analysis of the electric field intensity pattern has shown that the smallest focal spot size has been obtained when laser light propagated through ring gratings with alternating steps at a height $h_2$ equal to $2.65\lambda$ for a standard substrate (FWHM = $0.58\lambda$), which surpasses the results obtained using a diffractive axicon by 21.6%.

The longest light needle has been obtained for the case of a reverse annular grating at $S = 0.53\lambda$ for a direct GRIN substrate with DOF = $6.07\lambda$, which is 2.86 times longer than the light needle obtained using a diffractive axicon.

It has been demonstrated that the 3D microstructures in question make it possible to form both single optical traps and their sets. In particular, a narrow (FWHM = $0.82\lambda$) and an extended ($L = 1.52\lambda$) region of reduced intensity on the optical axis (optical trap) has been formed for an annular grating with alternating steps ($h_1 = 1.06\lambda$, $h_2 = 1.59\lambda$) with a standard substrate.

It has also been shown that when the reverse annular gratings are illuminated by the left-handed circularly polarized Laguerre–Gaussian (1,0) modes with $\sigma = 1.5$ μm, a set of optical traps with the FWHM from $0.25\lambda$ to $2.3\lambda$ and the length $L$ from $0.41\lambda$ to $1.1\lambda$ is formed on the optical axis.

**Author Contributions:** Conceptualization, D.A.S. and S.V.K.; methodology, S.V.K.; software, D.A.S.; validation, D.A.S.; formal analysis, S.V.K.; investigation, D.A.S.; resources, D.A.S. and S.V.K.; data curation, D.A.S. and S.V.K.; writing—original draft preparation, D.A.S.; writing—review and editing, D.A.S. and S.V.K.; visualization, D.A.S.; supervision, D.A.S.; project administration, D.A.S.; funding acquisition, D.A.S. and S.V.K. All authors have read and agreed to the published version of the manuscript.

**Funding:** This research was funded by the Ministry of Science and Higher Education by the scholarship of the President of the Russian Federation SP-1173.2022.5 in the parts «Introduction», «Materials and Methods», and «The formation of multiple optical traps by reverse ring gratings» and by the Samara University Development Program for 2021–2030 as part of the Priority 2030 program with the support of the Government of the Samara Region in the other parts.

**Institutional Review Board Statement:** Not applicable.

**Informed Consent Statement:** Not applicable.

**Data Availability Statement:** Not applicable.

**Conflicts of Interest:** The authors declare no conflict of interest.

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
