# Peer review of "Development of 3D Microstructures for the Formation of a Set of Optical Traps on the Optical Axis"

_photonics, doi:10.3390/photonics10020117_

Round 1

Reviewer 1 Report

The authors illustrate the diffraction of the circularly polarized LG(1,0) modes on GRIN substrate, including the standard, direct GRIN and reverse GRIN. Focusing features of the modes with different height of the relief rings are also simulated. Although this work is not that novelty and lack of any experimental confirmation, it might be practical for the groups who are working with optical traps formed by GRIN substrates.  I can not recommend the manuscript to be published on Photonics until all the following issues could be fixed:

1 I find it very confusing that the authors are not using color figures for all diffraction patterns. The present figures have very poor readability. I strongly recommend the authors to replace all the figures with color ones.

2 Lots of typos in the manuscript: line 33, line 72-73...

3 Since the manuscript is aiming at building a set of optical traps on the optical axis, only the diffraction pattern near focus point is not sufficient, the following simulation should be added:

  3.1 the optical force on typical sized particles should be illustrated;

  3.2 a set of optical traps on the axis are created, but once the first one traps a particle, can the following traps still trap particles? Or, how does this change the following traps?

Author Response

We are grateful to the reviewer for valuable comments that contributed to a significant improvement in the article, as well as to increase the significance and receptivity of the presented results. The answers are in the attached file.

Reviewer 2 Report

This paper has researched the focusing features of the LG modes through the FDTD method based on the GRIN or GRIN related materials. The results are very good, but it is a pity that no experimental results have been given. I think this paper can be accepted with a minor revision. Following are the suggestions.

1. The authors should check the English language clearly, and some spell mistakes should be corrected. In the Reference, "pow-er" should be "power";

2.Some references about the optical vortex and the optical tweezers should be cited.Zhang YX, Liu XF, Lin H, Wang D, Cao ES et al. Ultrafast multi-target control of tightly focused light fields. Opto-Electron Adv 5, 210026 (2022). doi: 10.29026/oea.2022.210026;Liu J, Zheng M, Xiong ZJ, Li ZY. 3D dynamic motion of a dielectric micro-sphere within optical tweezers. Opto-Electron Adv 4, 200015 (2021).. doi: 10.29026/oea.2021.200015;Dittrich S, Barcikowski S, Gökce B. Plasma and nanoparticle shielding during pulsed laser ablation in liquids cause ablation efficiency decrease. Opto-Electron Adv 4, 200072 (2021).. doi: 10.29026/oea.2021.200072.

Author Response

(The authors gave the same response as above.)

Reviewer 3 Report

Here, the authors proposed the use of graded-index (GRIN) substrate for 3D optical trapping. This concept can be interesting, however, the motivation needs to be better justified and the entire paper needs improvement before publication.

First, in the introduction section, the limitations of current optical tweezers technique and other variations need to be discussed in more details. For example, the high optical power and potential optothermal damages are recently overcome by Nature Nanotechnology, 2020, 908-913 and Science Advances 2021, 7, eabh1101. Other interesting variations include optoelectronic tweezers (Nature Photonics 5, 322–324 (2011)), optothermal tweezers (Chem. Rev. 2022, 122, 3, 3122–3179), and optical manipulation on solid substrates (Nature Communications 2019, 10, 5672). These recent advances should be discussed.

Second, the advantages of GRIN should be elaborated in more details. Especially, there are a number of advanced techniques for versatile manipulation, such as holographic tweezers (Nano Lett. 2017, 17, 7920-7925.) and DMD based optical manipulation (ACS Nano 2019, 13, 4, 3783–3795). More comparisons and discussions are suggested.

Third, what is the effect of topographical surface of the GRIN structure? I expect the heigh difference can lead to mechanical trap of objects. This effect needs to be mentioned and discussed.

Last, any application for chiral sensing or chiral light manipulation (Nano Lett. 2021, 21, 2, 973–979) since the circularly polarized beams are used?

Author Response

(The authors gave the same response as above.)

Round 2

Reviewer 1 Report

I'm satisfied with the revised version. The manuscript is suitable for publication on Photonics.